# Amniotic Mesenchymal-Derived Extracellular Vesicles and Their Role in the Prevention of Persistent Post-Breeding Induced Endometritis

**DOI:** 10.3390/ijms24065166

**Published:** 2023-03-08

**Authors:** Anna Lange-Consiglio, Giulia Gaspari, Federico Funghi, Emanuele Capra, Marina Cretich, Roberto Frigerio, Giampaolo Bosi, Fausto Cremonesi

**Affiliations:** 1Department of Veterinary Medicine and Animal Science (DIVAS), Università degli Studi di Milano, Via dell’Università, 6, 26900 Lodi, Italy; 2Independent Researcher, 58100 Grosseto, Italy; 3Istituto di Biologia e Biotecnologia Agraria (IBBA), Consiglio Nazionale delle Ricerche (CNR), 26900 Lodi, Italy; 4Istituto di Scienze e Tecnologie Chimiche “Giulio Natta” (SCITEC), Consiglio Nazionale delle Ricerche (CNR), 20133 Milan, Italy

**Keywords:** mare, endometritis, uterine fluid accumulation, polymorphonuclear neutrophil infiltration, regenerative medicine

## Abstract

Persistent post-breeding induced endometritis (PPBIE) is considered a major cause of subfertility in mares. It consists of persistent or delayed uterine inflammation in susceptible mares. There are many options for the treatment of PPBIE, but in this study, a novel approach aimed at preventing the onset of PPBIE was investigated. Stallion semen was supplemented with extracellular vesicles derived from amniotic mesenchymal stromal cells (AMSC-EVs) at the time of insemination to prevent or limit the development of PPBIE. Before use in mares, a dose–response curve was produced to evaluate the effect of AMSC-EVs on spermatozoa, and an optimal concentration of 400 × 10^6^ EVs with 10 × 10^6^ spermatozoa/mL was identified. At this concentration, sperm mobility parameters were not negatively affected. Sixteen susceptible mares were enrolled and inseminated with semen (n = 8; control group) or with semen supplemented with EVs (n = 8; EV group). The supplementation of AMSC-EVs to semen resulted in a reduction in polymorphonuclear neutrophil (PMN) infiltration as well as intrauterine fluid accumulation (IUF; *p* < 0.05). There was a significant reduction in intrauterine cytokine levels (*p* < 0.05) for TNF-α and IL-6 and an increase in anti-inflammatory IL-10 in mares in the EV group, suggesting successful modulation of the post-insemination inflammatory response. This procedure may be useful for mares susceptible to PPBIE.

## 1. Introduction

Persistent post-breeding-induced endometritis (PPBIE) is a major cause of subfertility in mares [1]. It is ranked as the third most commonly occurring medical problem in adult horses [2] and has an incidence of 15% in a normal population of Thoroughbreds [3]. Since it is associated with decreased fertility, it is a major concern for breeders and veterinary practitioners [4] and has significant economic implications.

Post-breeding endometritis is an inflammatory response in the uterus of a mare, triggered by exposure to semen. A transient, physiologically induced inflammatory response is seen in all mares after insemination, and it usually resolves spontaneously within 48 h. This inflammation clears from the uterus particles originating from the mating process, including debris, bacteria and semen, to establish a welcoming environment for embryo implantation. In mares considered susceptible to PPBIE, this inflammation persists and may progress to infectious endometritis or lead to endometrial fibrosis [5], with a reduction in pregnancy rates. Compared to susceptible mares, 6 h after sperm challenge, resistant mares showed higher mRNA expression for the anti-inflammatory cytokines IL-10, IL-1RN and IL-6 [6]. Interleukin-6 is a pleiotropic cytokine that promotes inflammation during the initial part of the immune response, while later it acts as an anti-inflammatory mediator. This dissimilarity between resistant and susceptible mares suggests a difference in the initial phase of the immune response against insemination and may help explain the transient nature of the inflammation in resistant mares, in contrast to the persistent inflammation in susceptible mares [6]. Mares prone to developing PPBIE have higher endometrial levels of pro-inflammatory cytokines IL-1b, IL-6, TNF-α and IL-8 compared to resistant mares even before mating [7]. The unbalanced and prolonged inflammatory condition within their uterus, caused by the prolonged expression of pro-inflammatory cytokines, might play an important role in the pathogenesis of persistent endometritis [8].

Current therapies focus on reducing the inflammation through the administration of ecbolic agents, anti-inflammatory drugs and antibiotics, often combined with uterine lavages. However, these treatments often fail to achieve a complete resolution of the pathological condition, probably because the process has already been triggered when they are administered. For this reason, novel alternative therapies are now being investigated, and the use of regenerative medicine seems promising. It has been shown that platelet-rich plasma (PRP) is able to modulate the uterine inflammatory response to semen in mares that are resistant to or susceptible to post-breeding endometritis [9,10]. Intrauterine infusion of autologous PRP is associated with a lower incidence of delayed uterine clearance following insemination and with improved pregnancy rates [11,12,13,14,15].

Mesenchymal stem cells (MSCs) have also been investigated for the treatment of uterine pathologies with respect to their antiapoptotic, chemotactic and immune-modulatory properties. Human bone marrow-derived mesenchymal cell (hBM-MSCs) transplants have been able to promote endometrial tissue regeneration and enhance fertility rates in mouse models [16,17]. In the field of PPBIE treatment, Ferris et al. [18] showed that intrauterine MSCs administration reduced the number of neutrophils in the uterine lumen and increased IL-1Ra expression in the endometrium of normal mares.

Amniotic mesenchymal stromal cells (AMSCs; nomenclature defined by Silini et al., 2020 [19]) possess strong immunomodulatory properties that promote tissue repair and the restoration of tissue homeostasis [20,21]. Their effects have been successfully investigated in many in vitro [22] and in vivo preclinical studies, often for the treatment of orthopedic disorders in many animal models, including rats [23,24,25], rabbits [26] and horses [27]. AMSCs have also been reported to support equine endometrial regeneration in vitro [28], suggesting their possible application as a therapy for uterine pathologies.

However, only a small percentage of transplanted MSCs are able to engraft and survive in the inhospitable and inflammatory environment in the presence of disease [29]. Indeed, the differentiation of transplanted MSCs into damaged tissue seems to contribute little to their therapeutic benefits [30]. Recent data suggest that the action of transplanted MSCs might involve the secretion of mediators such as cytokines, chemokines and growth factors that could drive immunomodulation via paracrine signaling. Conditioned medium (CM), secreted by cells during culture, has been shown to promote the structural and functional regeneration of renal [31], lung [32], cardiac [33,34] and tendon [27] tissues. Corradetti et al. (2014) demonstrated that AMSC-conditioned medium (AMSC-CM) was effective in the replenishment of endometrial cells and uterine regeneration [28]. To date, only one study has evaluated the in vivo effects of MSC-CM intrauterine infusion on mares susceptible to PPBIE, showing the effective downregulation of pro-inflammatory IL-6 and the upregulation of anti-inflammatory IL-10 in the uterine endometrium, as well as the reduction in PMN infiltration 6 h post-insemination [35].

In addition to soluble factors, CM is also composed of an insoluble fraction, represented by extracellular vesicles (EVs). These double-layered vesicles, originating from the plasma membrane of cells, are secreted by cells and represent an integral part of the intercellular environment, acting as regulators of cell-to-cell communication. The vehiculation of active molecules, such as lipids, proteins, DNAs, mRNAs and microRNAs (miRNAs), to their target cells allows them to act as signaling complexes by promoting proliferation and angiogenesis, preventing apoptosis, modulating the immune system, and suppressing fibrosis [36]. It has been demonstrated that, since they are able to differentially modulate the function of T, B and natural killer cells, EVs isolated from any cell source have immunological properties [37,38]. The incorporation of EVs derived from AMSCs (AMSC-EVs) into endometrial cells helps counteract the inflammatory action of LPS, both by down-regulating the levels of pro-inflammatory factors and by up-regulating anti-inflammatory factors secreted by cells [39]. In addition, EVs are considered key players in many stages of gamete maturation [40,41,42,43,44], fertilization [42,45,46,47,48] and implantation [49,50,51,52,53]. They are involved in mutual paracrine communication between embryonic and maternal environments at the early stages of pre-implantation and embryonic development [54,55]. In vivo intrauterine AMSC-EV administration resulted in the resolution of the inflammatory condition of a mare affected by chronic degenerative endometritis and restored the correct fetal–maternal crosstalk, leading to a successful pregnancy and the histological improvement of the pathological condition [56].

Given this evidence, the aim of this study was to evaluate an alternative approach aimed at preventing the onset of PPBIE by combining AMSC-EVs with semen at the time of artificial insemination. The hypothesis is that AMSC-EVs might be able to rebalance the altered cytokine profile of mares prone to developing PPBIE and modulate the inflammatory response triggered by spermatozoa. The study also sought to confirm that exposure to AMSC-EVs did not alter sperm vitality and mobility by assessing motility parameters.

## 2. Results

### 2.1. Tissue Collection, Amniotic Mesenchymal Stromal Cell Isolation and Expansion

Cells isolated from the amniotic membrane showed 90% viability after Trypan blue exclusion evaluation. They were able to adhere to the plastic surface of culture flasks and had a fibroblast-like morphology. These cells were expanded at passage three, and the standardized protocol used to obtain these cells allowed the isolation of AMSCs expressing CD29, CD44, CD106, CD105 and MHC-I surface markers, but not CD34 and MHC-II. These cells also have the potential to differentiate into mesenchymal (osteogenic, adipogenic and chondrogenic) and ectodermic (neurogenic) lines [57].

### 2.2. Production of CM, Isolation and Characterization of Isolated AMSC-EVs (Via Nanosight Dot Blot, Western Blot Analysis and Transmission Electron Microscopy)

At passage three, CM was collected, and AMSC-EVs were isolated through ultracentrifugation and characterized according to MISEV guidelines [58]. Their concentration and dimensions were evaluated via nanoparticle tracking analysis with the NanoSight LM10 instrument. The NanoSight analysis revealed an average size of 257.7 ± 11.4 nm and a concentration of 1.26 × 10^11^ particles/mL. Given the obtained dimensions, the pool of isolated EVs was composed predominantly of microvesicles (Figure 1A).

Western blot analysis allowed verification of the presence of specific EVs’ internal markers, Alix and TSG101 (Figure 1B). According to dot blot analysis, specific surface markers, CD9 and CD81, were also expressed by the isolated vesicles (Figure 1C), confirming that the preparation contained EVs.

Transmission electron microscopy (TEM) confirmed the efficiency of the isolation method for AMSC-EV, as revealed by the spheroid morphology of the EVs, with a moderately electron-dense coat (Figure 1D). The ultrastructure of the AMSCs shows EVs near the membrane of the cell. Rarely, multivesicular bodies were detected, meaning that AMSCs produce few exosomes.

### 2.3. Effects of Exposure to AMSC-EVs on Motility of Semen

To evaluate whether exposure to AMSC-EVs could alter sperm motility parameters, a dose–response curve was carried out, exposing spermatozoa to different concentrations of AMSC-EVs for increasing incubation periods. Then, sperm motility values were assessed with a computer-assisted semen analysis (CASA) system. The percentages of motile and progressive spermatozoa and sperm motility parameters after co-incubation with AMSC-EVs are shown in Table 1. Unexpectedly, with increasing EV concentrations and testing times, no linear increase in effects was observed. However, 400 × 10^6^ EVs/mL was considered the most appropriate concentration of all the evaluated conditions. Indeed, at this concentration, sperm motility and progressivity parameters were improved compared to the controls at all time points (Table 1). VCL values remained similar for all EV concentrations for the first hour of incubation; in the following hours, VCL decreased except for the concentration of 400 × 10^6^ EVs/mL, for which VCL values were similar to the control. The VSL was similar to the control values only at a concentration of 400 × 10^6^ EVs/mL (Appendix A). Similarly, AHL (which detects the movement of the head of spermatozoa, indicating a wave motion) was unaltered at this concentration compared to the control for all the time points. Interestingly, at the concentration of 400 × 10^6^ EVs/mL, the linearity parameter (LIN) increased after the first and second hours and remained statistically higher throughout the incubation period compared to the control. In the same way, STR values did not differ from the control at the first and third hours, while they rose during the second and fourth hours (Appendix A). Conversely, since it caused an enhancement of the lateral movement of the head (the AHL parameter increased), resulting in a reduction in LIN and STR values, 300 × 10^6^ EVs/mL was considered the worst concentration tested.

### 2.4. AMSC-EVs Labeling and Uptake Evaluation

After EV labeling with PKH-26, the semen was exposed to 400 × 10^6^ AMSC-EVs/mL. The co-culture of AMSC-EVs with spermatozoa resulted in an improvement in semen motility values. We, therefore, investigated whether spermatozoa were able to incorporate AMSC-EVs at this concentration. The spermatozoa and AMSC-EVs were labeled with Hoechst 33342 and PKH-26, respectively, and the uptake was confirmed using confocal microscopy. The fluorescence remained stable during all four hours of incubation and showed the incorporation of AMSC-EVs at the level of the intermediate tract of the spermatozoa (Figure 2A–C). Through TEM, it was possible to observe that the spermatozoa membrane remains intact after EV incorporation (Figure 2D).

### 2.5. In Vivo Anti-Inflammatory Effects of AMSC-EV Supplementation of Inseminating Dose in Susceptible Mares

#### 2.5.1. Evaluation of Semen Used for AI

The spermiogram of the stallion selected for AI is summarized in Table 2.

The ejaculate showed hypozoospermia and hypospermia. The progressive and rectilinear motility were below the limits indicated for this species. The percentages of viable and normal (without anomalies) spermatozoa were within the standard ranges.

#### 2.5.2. Polymorphonuclear Neutrophil Infiltration

Polymorphonuclear neutrophil infiltration was evaluated as an indicator of the inflammatory status of the uterine environment. The susceptible mares used as controls showed a higher PMN count at 6 and 24 h post-insemination compared to the mares treated with AMSC-EVs (*p* > 0.05) (Figure 3 and Table 3).

#### 2.5.3. Intrauterine Fluid Accumulation

The total intrauterine fluid accumulation (IUF) was measured to assess the inflammatory condition of the uteri. Ultrasound evaluation revealed a reduction in IUF accumulation after AMSC-EVs treatment, while the control mares had higher volumes of fluid, both at 6 and 24 h post-insemination (*p* < 005) (Table 4). 

#### 2.5.4. Cytokine Evaluation

To assess the efficacy of the in vivo approach, the levels of pro- and anti-inflammatory cytokines secreted into the uterine fluid were evaluated with an ELISA assay. There was no statistical difference in the concentration of TNF-α among times or groups (*p* > 0.05) (Figure 4A,B). In the control group, IL-6 concentration increased considerably 6 h post-insemination and gradually decreased until 24 h (Figure 4C). The AMSC-EV treatment resulted in a reduction in the levels of IL-6 compared to the control (Figure 4D), and these remained constant until 24 h post-insemination (Figure 4C).

After insemination, anti-inflammatory IL-10 release into the IUF remained constant and did not differ significantly from the baseline (0 h) in the control group. In contrast, after AMSC-EV treatment, IL-10 levels significantly increased 6 h post-insemination and remained almost unaltered until 24 h (Figure 4E). For all the time points tested, AMSC-EV treatment resulted in a higher expression of anti-inflammatory IL-10 compared to the control group (Figure 4F).

The fertility rate was similar between the CTR group and EV group (*p* > 0.05; control: 62.5%, 5/8; EV group: 87.5%, 7/8).

## 3. Discussion

In this study, the efficacy of a novel treatment approach based on the addition of AMSC-EVs to the inseminating dose for the prevention of the onset of PPBIE in susceptible mares was evaluated.

In these mares, as sperm reach the uterus, the local innate immune response is activated by the recognition of the antigen (spermatozoon) by Toll-like receptors (TLRs). This event starts the inflammatory cascade that triggers the activation of the downstream nuclear factor-kbeta (NF-kB). The NFkB pathway activates genes encoding pro-inflammatory cytokines, chemokines and cyclooxygenase-2 (COX-2) [59]. In the equine endometrium, COX-2 is expressed after insemination [60]; moreover, COX-2 induces a local endometrial increase in PGF 16 h after insemination [61]. The activation of cytokines, chemokines and the complement cascade recruit leukocytes and immune cells to the site of inflammation to allow the phagocytosis of sperm and bacteria. Equine spermatozoa also activate the complement cascade, leading to an increase in complement factors C3b and C5a, leukotrienes and prostaglandins (PGs), thus resulting in the chemotaxis of PMNs in the uterus. Pro-inflammatory cytokines increase uterine vascular permeability [59] favoring the formation of a transudate that causes edema and fluid accumulation. Permeability alterations lead to the expression of adhesion molecules by endothelial cells and leukocyte diapedesis. The expression of P- and L-selectins induces neutrophil chemotaxis. Neutrophils are fundamental for sperm and bacteria phagocytosis, but they also secrete additional cytokines and chemoattractant mediators, further contributing to inflammation [59]. Leukocytes then release PG, which promotes myometrial contractility [62].

The development of PPBIE In susceptible mares compromises their fertility as it makes the uterine environment inhospitable for embryo implantation and development.

During the final part of the immune process, it is crucial that the appropriate balance between pro- and anti-inflammatory cytokines is re-established. The approach proposed in this study aims at favoring the restoration of endometrial physiological homeostasis after sperm challenge, thus preventing the persistence of inflammation.

Previous studies have shown how the intrauterine administration of MSC-CM reduces insemination-induced inflammation both in resistant [18] and susceptible mares [35]. After this treatment, the concentrations of anti-inflammatory IL-10 increased, while the levels of pro-inflammatory IL-6 decreased. This suggests that paracrine communication mechanisms might play a consistent role in immunomodulation.

Since EVs are an integral part of CM, their immunomodulatory contribution to the equine uterine environment was investigated. It was demonstrated that AMSC-EVs possess proliferative, anti-apoptotic, pro-angiogenic, anti-fibrotic and immunomodulatory properties [36] in different tissues. In vitro, epithelial endometrial cells can incorporate AMSC-EVs, which transport a variety of molecules (such as proteins, mRNA and miRNA) necessary for both the secretion of anti-inflammatory factors and the reduction in pro-inflammatory elements, into target cells [39]. A successful pregnancy and a progressive reduction in histological changes in the endometrial tissue of a mare affected by chronic degenerative endometritis were detected in vivo after cyclical AMSC-EV administrations [56]. These data offer hints as to the potential involvement of AMSC-EVs in the effective mitigation of endometrial inflammation induced by mating.

Although the primary aim of this study was to investigate the effect of supplementing sperm with EVs for artificial insemination, it was important to ensure there would be no deleterious effects on the spermatozoa. Thus, at first, the effects of these nanoparticles on sperm motility parameters were evaluated to ensure the supplementation would not impair the fertilization process.

Physiologically, spermatozoa also communicate with surrounding tissues through the paracrine exchange of EVs during their transit along both female and male reproductive tracts. Consequently, it can be speculated that spermatozoa could also interact with AMSC-EVs. To verify this hypothesis, fresh semen was exposed to increasing concentrations of AMSC-EVs and monitored every hour for 4 h. It has been estimated that it takes four hours for most spermatozoa to reach the fallopian tubes after insemination [63]. So, it is likely that, in vivo, spermatozoa will rapidly leave their dilution medium on entering the female genital tract and will thus be in contact with AMSC-EVs in a much shorter time than the period tested.

Analysis via the CASA system of the semen used for the in vitro study showed motility parameters below the average ranges of equine species. However, it is known that sometimes there is no correspondence between the quality of the semen evaluated in vitro and the reproductive performance of the animal. Indeed, an excellent in vitro quality of an ejaculate does not necessarily correspond to the in vivo prolificacy of the animal and, conversely, a mediocre-quality ejaculate may have high prolificacy in vivo. Fertilization is a multi-factorial process depending not only on spermatozoa but also on their interaction with the various levels of the female genital tract and the oocyte itself [64]. The stallion used for AI in this study was chosen by the breeder precisely for his proven fertility, even if its semen demonstrated poor quality in vitro. Indeed, this semen was always used fresh.

After co-incubation with AMSC-EVs, analysis via the CASA system did not detect any alteration in sperm mobility parameters. Of all the concentrations tested, 400 × 10^6^ EVs/mL was the only one at which the motility and progressivity percentages were unaltered or even improved. VCL also remained constant throughout the four-hour incubation period and was not different from the control. A similar finding for the AHL parameter and for VSL was shown until the third hour. Interestingly, LIN values increased after the first and second hours and STR parameters at the second and fourth hours.

These observations allowed us to verify whether spermatozoa, in addition to not being vulnerable to AMSC-EV exposure, were also able to incorporate AMSC-EVs. Confocal microscopy images confirmed the uptake of AMSC-EVs at a concentration of 400 × 10^6^ EVs/mL and showed that spermatozoa stably incorporate EVs at the level of the intermediate tract. This result might explain the improvement in some of the mobility parameters observed, meaning that AMSC-EVs might carry some ribonucleic or protein molecules or miRNAs into the spermatozoa, enabling a functional modification. 

Since in vitro testing confirmed that 400 × 10^6^ EVs for 10 × 10^6^ spermatozoa was a safe concentration, it was chosen for the insemination of the mares assigned to the treatment group. For in vivo treatment, an amount of 20 × 10^9^ EVs was estimated, considering an inseminating dose of 500 × 10^6^ spermatozoa regardless of the final dilution used for AI, which is different compared to the in vitro study.

The development of the inflammatory response was monitored at 0, 6, 12, 18 and 24 h post-insemination by evaluating the levels of cytokines released into the uterine fluid. These time points were chosen because, in susceptible mares, usually, neutrophils are detected in the uterine lumen within 30 min following artificial insemination and peak between 6 and 12 h. In addition, the expressions of TNF-α and IL-6 reach their highest levels at 2 h and 6 h post-insemination, respectively. Susceptible mares present lower expression of modulatory cytokines IL-10 at 6 h compared to resistant mares [65]. Based on these dynamics, the mares were monitored only until 24 h and not over this time.

In our study, there was no significant difference in the TNF-α concentration between the two groups through ELISA analysis. The IL-6 concentration notably increased at 6 h post-insemination in the control group. In normal mares, pro-inflammatory cytokines prevail until the sixth hour, and IL-6 participates in the activation of inflammation. From the sixth hour onwards, resistant mares start expressing anti-inflammatory cytokines and IL-6, which physiologically has a pleiotropic nature with anti-inflammatory features. In the treatment group, IL-6 levels were significantly lower after the sixth hour post-insemination and remained stable for up to 24 h. This supports the hypothesis that the inflammatory balance is successfully restored by AMSC-EVs.

IL-10 was the only anti-inflammatory cytokine studied. In the control group, the levels of IL-10 remained almost unchanged throughout the study period. This highlights how, in susceptible mares, the anti-inflammatory response failed to counteract the pro-inflammatory mediators. However, following treatment with AMSC-EVs, the levels of IL-10 increased significantly from six hours after insemination.

The observed effects are probably due to vesicle internalization by the endometrial epithelial cells of the treated mares, as supported by the in vitro study conducted by Perrini et al. [39]. The reported data are also consistent with the results of Tongu et al. [35] on MSC-CM.

Extracellular vesicles represent the insoluble fraction of the stem cell secretome, and, according to our data, they seem not only to play an active role in immunomodulation mechanisms, but they may also contain the minimal molecular factors required for immunosuppression. It has recently been demonstrated that AMSC-EVs are able to convey immunoregulatory information. Through the vehiculation of specific miRNAs, they interfere with the modulation of interleukin signaling and with inflammatory processes. Indeed, Lange Consiglio et al. investigated the miRNA cargo of equine AMSC-EVs, revealing strict compartmentalization of specific miRNAs into the secreted vesicles. Three molecules were particularly enriched in the EV population: miR-223, miR-150 and miR-126, which were involved in the down-regulation of pro-inflammatory cytokine expression by macrophages in the promotion of angiogenesis and inhibition of inflammation in endometrial cells [38]. This work not only matches our observations but also supports our hypothesis of EVs as selected effectors of immune regulation. In this context, another possible use of AMSC-EVs could be a pre-breeding intrauterine infusion as an alternative treatment. Indeed, AMSC-EVs could be internalized in endometrial cells, as evidenced in in vitro studies [39], and transfer these specific miRNAs, inhibiting the synthesis of protein molecules such as proinflammatory factors responsible for persistent inflammation.

Furthermore, since AMSC-EVs derive from a membrane of the placental adnexa, they are physiologically part of the interface between fetal and maternal tissues and contribute to the maintenance of fetal tolerance.

Despite the significant improvement in post-insemination inflammation through semen supplementation with ACM-EVs, there were no statistically significant differences in pregnancy rates between the CTR and AMSC-EVs groups. We assume that the limited number of susceptible mares enrolled in the present study does not allow for the establishment of statistical differences. However, further studies that include a larger number of individuals, as well as resistant mares, are required to ensure statistical relevance. It would also be interesting to test for a larger number of cytokines.

Another limitation of this study is the lack of crossovers between the two experimental groups, intended for the use of the treated mares as a control group and vice versa. The mares enrolled in this study are not experimental animals but are privately owned. For this reason, after insemination, the pregnancy will be monitored until delivery, without the possibility of crossing the two groups.

Nevertheless, these preliminary results encourage the use of this therapy in clinical practice for the treatment of PPBIE-susceptible mares. This treatment could prevent the onset of PPBIE and avoid the alteration of the inflamed endometrium into fibrotic tissue, which is more susceptible to developing persistent endometritis and resulting in infertility [65].

## 4. Materials and Methods

### 4.1. Materials and Animals

All chemicals in this study were purchased from Sigma-Aldrich (Milan, Italy) unless otherwise stated, whereas disposable materials were purchased from Euroclone (Milan, Italy).

Mare placentas (n = 3) were collected from three broodmares at the term of a normal pregnancy. All procedures were conducted following standard veterinary practice and in accordance with the 2010/63 EU directive on animal protection.

Fresh semen for in vitro study was collected by means of an artificial vagina from three adult stallions of proven fertility belonging to the Equicenter Clinic of Monteleone (Pavia, Italy). All collections were performed according to approved animal care procedures.

Mares enrolled in this study were made available by private owners after informed consent. The study was conducted with the approval of the Ethics Committee of UNIMI (protocol code OPBA_118-2017).

### 4.2. Experimental Design

This study was carried out both in vitro and in vivo.

In vitro studies: (i) isolation of AMSCs by enzymatic digestions; (ii) collection of CM and isolation of AMSC-EVs that were analyzed for concentration and size through Nanosight, for external and internal membrane antigens through Western blot and dot blot, and for morphology through TEM; (iii) labeling of AMSC-EVs with PHH-26 and study of dose–response curve by motility parameters with CASA system; (iv) evaluation of AMSC-EV incorporation in equine spermatozoa using a confocal microscope.

In vivo study: (i) selection of stallion for in vitro study and in vivo insemination; (ii) selection of susceptible mares; (iii) evaluation of PMN on cytobrush, IUF accumulation and cytokine, before and after insemination.

### 4.3. Tissue Collection and Amniotic Mesenchymal Stromal Cell Isolation

Three allanto-amniotic membranes were obtained at the term of a normal and physiological pregnancy, transported at 4 °C in phosphate-buffered saline solution (PBS) supplemented with 4 mg/mL amphotericin B (Euroclone), 100 U/mL penicillin (Euroclone) and 100 mg/mL streptomycin (Euroclone), and were processed within 8 h of collection. Amniotic membranes were mechanically separated from the overlaying allantois and cut into small sections of 9 cm^2^ each, for a total weight of approximately 12 g and an extension of 630 cm^2^. Then, amniotic fragments were incubated for 9 min at 38.5 °C in PBS containing 2.4 U/mL dispase (Becton Dickinson & Company, Milan, Italy). After a final incubation of 5–10 min at room temperature in HG-DMEM (high-glucose Dulbecco’s modified Eagle’s medium, Euroclone) supplemented with 10% fetal bovine serum (FBS) and 2 mM L-glutamine, fragments were digested with 1 mg/mL collagenase type I and 20 µg/mL DNase (Roche, Mannheim, Germany) for 3 h at 38.5 °C.

Fragments were then removed, and the products of enzymatic digestion were passed through a 100 µm filter. Dissociated cells were collected through centrifugation at 250× *g* for 10 min.

Isolated cells were defined as AMSCs.

#### 4.3.1. AMSCs Expansion

Isolated AMSCs were seeded at a density of 1 × 10^5^ cell/cm^2^ for the first passage and at a density of 1 × 10^4^ cell/cm^2^ for subsequent passages in HG-DMEM enriched with 10% FBS, penicillin (100 UI/mL)–streptomycin (100 UI/mL), 0.25 mg/mL amphotericin B, 2 mM L-glutamine and 10 ng/mL epidermal growth factor (EGF). Cellular cultures were kept in an incubator at controlled atmosphere with 5% CO_2_, 90% humidity and 38.5 °C temperature until passage 3 (P3).

Cellular vitality was measured via the Trypan blue exclusion method.

#### 4.3.2. CM Production

AMSCs at P3 were maintained in serum-free Ultraculture medium (Ultraculture, Lonza, Milan, Italy), in a controlled atmosphere with 90% humidity, 5% CO_2_ and 38.5 °C, for 3 days. Conditioned media were collected each morning from the culture flasks and replaced with fresh media. Culture media were collected and centrifuged at 1600× *g* for 20 min to discard cells and at 4500× *g* for 20 min to discard debris, and then stored at −20 °C until EV isolation.

### 4.4. Isolation of Extracellular Vesicles

The obtained CM was ultracentrifuged at 100,000× *g* (Beckman Coulter OptimaX, Milan, Italy), 4 °C for 1 h. The pellet was resuspended in a serum-free medium, and an aliquot was collected for EV concentration and dimension measurements. After Nanosight results were obtained, EV aliquots were prepared and stored at −20 °C with 1% dimethylsulfoxide until following steps. For the subsequent experiments, different EV aliquots were thawed in a water bath at 38.5 °C.

#### Characterization of Isolated AMSC-EVs (Via Nanosight, Dot Blot, Western Blot Analysis and Transmission Electron Microscopy)

The isolated AMSC-EVs were characterized following MISEV guidelines [58]. EV size and concentration parameters were obtained using nanoparticle tracking analysis (NTA) performed according to manufacturer’s instructions using a NanoSight NS300 system (Malvern Technologies, Malvern, UK) configured with 532 nm laser. All samples were diluted in filtered PBS to a final volume of 1 mL. We performed several dilutions to identify the ideal condition (20–100 particles must be found in each frame). The following settings were used according to the manufacturer’s software manual. A syringe pump with constant flow injection was used, and three videos of 60 s each were captured and analyzed with NTA software version 3.4. From each video, the mean, mode and median EV sizes were used to calculate sample concentration expressed in nanoparticles/mL.

EV markers were assessed via Western blotting. An amount of 32 µL of each sample was treated with 8 µL of Laeimmli buffer in redaction condition, and the preparation was heated for 10 min at 95 °C. The sample was loaded on electrophoretic gel SDS-PAGE (4–20%, Mini-Protean TGX Precast protein gel, Bio-Rad) and run under an electric field before being transferred into the nitrocellulose membrane (BioRad, Trans-Blot Turbo, Milan, Italy). A blocking step was performed to saturate nonspecific sites: 1 h with 5% (*w*/*v*) BSA in T-TBS (tris-buffered saline: 150 mM NaCl, 20 mM Tris-HCl, pH 7.4 and 0.5% Tween 20). After, the membranes were incubated overnight at 4 °C on an orbital shaker with primary antibodies, polyclonal antibody anti-TSG101 (1:500 dilution, Invitrogen, Monza, Italy) and monoclonal antibody anti-Alix (1:1000 dilution, Santa Cruz, CA, USA). Strips were washed for 5 min, 3 times with TBS-T, after which membranes were incubated with horseradish peroxidase-conjugated, anti-mouse secondary antibody (BioRad) diluted 1:3000 in TBS-T with 1% BSA. Final washes were performed, and the signal was detected using Bio-Rad Clarity Western ECL Substrate (Bio-Rad) and imaged using a Chemidoc XRS+ (BioRad).

Dot blot analysis was used to detect the EV surface markers, as previously described (Lange-Consiglio et al. Theriogenology, 2022). CD9 and CD81 were detected using mAb anti-CD9 (1:500 dilution; Bio-Rad) and mAb anti-CD81 (1:500 dilution; AbbexaLLC, Abcam, Cambridge, UK) as primary antibodies, and they were revealed by an HRP-secondary antibody. All membranes were read using Bio-Rad Clarity Western ECL Substrate (Bio-Rad) and imaged using a Chemidoc XRS+ (BioRad).

Transmission electron microscopy (TEM) was carried out as briefly described: a drop of 10 µL EVs, with a concentration of 20 × 10^9^ mg/mL, was absorbed on 300-mesh formvar/carbon copper grids. Extracellular vesicles were then fixed with a solution containing 2.5% glutaraldehyde for 5 min. After repeated washings in distilled water, the grids were contrasted with 2% uranyl acetate, air-dried, and examined using a transmission electron microscope [66]. The same protocol was used for TEM analysis of spermatozoa. For TEM detection of EVs secreted by cells, AMCs were processed as previously described by Lange-Consiglio et al. [67].

Digital images were acquired.

### 4.5. AMSC-EV Labeling

To track the EV uptake by spermatozoa through confocal microscopy, EVs were labeled with PKH-26, a red aliphatic fluorescence chromophore that can intercalate into the lipid bilayer. EVs were ultracentrifuged, and the resulting pellet was diluted to 1 mL with the reagent supplied by the kit. An amount of 4 µL of fluorochrome was added, and the suspension was incubated in the dark for 30 min, 38.5 °C, 5% CO_2_. The reaction was stopped by adding 7 mL of serum-free DMEM. The suspension was ultracentrifuged again at 100,000× *g* at 4 °C for 1 h, and the final pellet was resuspended in HG-DMEM and stored at −20 °C until use.

### 4.6. Effects of Exposure of AMSC-EVs on Motility Parameters of Semen

Fresh semen from three stallions was obtained from the Equicenter Clinic of Monteleone (Pavia, Italy). Immediately after collection, sperm concentration was evaluated with a sperm count spectrophotometer (IMV Technology, Piacenza, Italy). Fresh semen from each stallion were pooled and diluted 1:1 in pre-heated INRA96 (IMV Technologies) at 38.5 °C with 5% CO_2_, and then centrifuged at 100× *g* for 1 min at 38.5 °C, to remove debris and non-viable spermatozoa. The resulting supernatant was transferred into round-bottom tubes and centrifuged at 600× *g* for 5 min at 38.5 °C. The pellet containing vital spermatozoa was resuspended in INRA with a concentration of 10 × 10^6^ spz/mL.

A dose–response curve was produced by exposing 1 mL of semen to increasing concentrations of EVs for different amounts of time. Specifically, concentrations ranging from 0 × 10^6^ EVs/mL (control) to 500 × 10^6^ EVs/mL were tested for 1 h, 2 h, 3 h and 4 h incubation periods. The experiment was performed in triplicate.

Each sample was evaluated using a computer-assisted semen analysis (CASA) system that allows a complete analysis of sperm motility parameters. It is composed of a computer PC Pentium IV^TM^, 1GB RAM; Windows XP Professional operating system with motility analysis software; high-resolution and sensitivity video camera (CFW 131-OM, Scion Corporation, Kaya Instruments, Nesher, Istrael) for morphological analysis; high-resolution and acquisition video camera (Basler, A6021-2, Kaya Instrumments) for motility analysis; SPERMOT IMMAGINI & COMPUTER software (Bareggio, Milan, Italy) for motility and morphological analysis; phase contrast microscope Olympus BX 51 with heated table (Lenham, MC60D); phototube and c-mount adapter with 2.5× magnification; 20× objective (Olympus PH/NHS2, 20× negative phase contrast) for motility analysis; 20× negative phase contrast objective (Olympus Achroplan 20×) for morphological analysis. 

For motility analysis, CELL VU (IMV, Technologies, Italy) slides with two 20 µm-depth chambers each were used. Each sample was loaded, keeping the slides at 37.5 °C throughout the analysis. For each sample, five randomly selected fields were registered, capturing 60 photograms per second.

### 4.7. AMSC-EV Uptake

Fresh semen was processed as previously described. One mL of semen with 10 × 10^6^ spz/mL concentration was exposed to 400 × 10^6^ EVs/mL for different incubation periods (1 h, 2 h, 3 h, 4 h). During the last 15 min of incubation at each time point, spermatozoa were labeled with Hoechst 33342 (10 µg/mL) and then centrifugated at 200× *g* for 5 min. The resulting pellet was resuspended in 50 µL PBS with 10 µL formalin to fix spermatozoa for confocal microscopic observation to detect the site of AMSC-EV uptake. A Leica SP2 laser scanning microscope (Leica Microsystem Srl, Milan, Italy) equipped with a dry PL Fluotar 20× AN 0.5 objective was used. 

### 4.8. In Vivo Anti-Inflammatory Effects of AMSC-EV Supplementation of Inseminating Dose in Susceptible Mares

#### 4.8.1. Semen Collection

A 10-year-old Quarter Horse stallion with known fertility who belonged to the Equicenter Clinic Monteleone (Pavia, Italy) was enrolled in this in vivo study.

After collection through an artificial vagina, fresh semen was filtered to discard debris and gel proteolytic fraction, and then diluted 1:3 with pre-heated INRA 96. Semen was kept at 38.5 °C for motility and concentration analysis; then, semen was diluted to obtain 500 × 10^6^ vital spermatozoa with progressive motility in 30 mL.

#### 4.8.2. Mare Selection and Insemination Combining Semen and EVs

Sixteen mares belonging to different private owners and previously characterized as susceptible to PPBIE (based on PMN counts and IUF accumulation 48 h after insemination) were enrolled in this study. These mares did not show histopathological lesions in the endometrium, but once monitored by ultrasound examination, they usually developed an intrauterine fluid (IUF) accumulation greater than 20 mm and a PMN infiltration higher than 20% 48 h after insemination with 500 × 10^6^ total spermatozoa. The detection of IUF 24–48 h after breeding suggests inadequate or delayed uterine clearance. Examination for the presence of IUF is the most used and most practical marker for susceptibility in practice and clinical field studies in diagnosing PPBIE [65].

For this study, 8 mares were randomly assigned to the treatment group (EVs), and another 8 mares were used as controls (CTR). After their estrous cycle, endometrial culture and cytological analysis were performed for the presence of aerobic bacteria. Both the experimental groups showed negative results (2% PNM count and negative bacterial culture).

Mares were monitored daily by transrectal ultrasonography. Once a 35 mm pre-ovulatory follicle with endometrial edema was identified, control mares were artificially inseminated with a standard procedure, while mares in the EV group were inseminated with a combination of semen and AMSC-EVs. The insemination dose was set at 500 × 10^6^ total spermatozoa, diluted in 30 mL of INRA96 and supplemented with 20 × 10^9^ AMSC-EVs, considering the concentration of 400 × 10^6^ vesicles per 10 × 10^6^/mL of semen used for the in vitro study.

Mares were examined 24- and 48-hours post-insemination to check for ovulation and the presence of IUF. The accumulation, height and width of IUF column were measured at the level of uterine bifurcation using an ultrasound caliper.

Exfoliative cytology samples were collected 0, 6, 12, 18 and 24 h after the insemination using a disposable cytobrush (IMV Technologies, Piacenza, Italy). Briefly, the mares’ perineum was cleansed with a gentle detergent, washed with clean water, and dried with absorbent paper to avoid contamination. The cytobrush was manually guided into the vagina and the cervix using a sterile sleeve. The brush was inserted deep into the uterus and delicately moved to collect endometrial cells. The fluid (~1.5 mL) was kept in the covered sheath of the cytobrush kit and transferred into a plastic tube. It was then centrifuged at 400× *g* for 20 min at 5 °C. After the first centrifugation, the supernatant was collected and centrifuged again at 2000× *g* for 20 min at 5 °C to remove cellular debris. Then, the fluid was snap-frozen in liquid nitrogen (−196 °C) for further analysis.

The obtained cytobrushes were smeared on microscope slides, air-dried and colored with Dip Quick (Securlab, Rome, Italy). Samples were microscopically examined with a 100× objective, and PNMs were counted. 

Twenty-four hours after insemination and collection of samples, mares were treated with a standardized procedure consisting of uterine lavages with 2 L of Ringer solution and oxytocin administration (20 units, i.m.), twice a day until 96 h post-insemination.

Mares were examined for pregnancy diagnosis 14 days after ovulation.

#### 4.8.3. Cytokine Evaluation

Uterine fluid samples collected 0-, 6-, 12-, 18- and 24-hours post-insemination were kept at room temperature. The levels of pro-inflammatory TNF-α were assessed via commercially available TNF-α ELISA Kit (Termo Fisher Scientific, Milan, Italy; catalog number ESS0017). Levels of IL-6 were assessed via commercially available Equine IL-6 DuoSet ELISA (R&D System, Bio-Techne, Milan, Italy; catalog number DY1886). Anti-inflammatory IL-10 cytokine was measured through commercially available equine ELISA Kit (Termo Fisher Scientific, Milan, Italy; catalog number EEIL10X5). All analyses were carried out following supplier’s instructions.

Ranges of the assays were: 15.6–1000 pg/mL for TNF-α; <125–8000 pg/mL for IL-6; and 0.122–25 ng/mL for IL-10.

### 4.9. Statistical Analysis

The obtained data were analyzed through the analysis of variance (ANOVA) test. Differences were considered statistically significant if *p* value was <0.05. Conception rates were evaluated using chi-square test. Statistical significance was set at *p* < 0.05. Statistical analysis was performed using GraphPad Instat 3.00 for Windows (GraphPad Software, La Jolla, CA, USA).

## 5. Conclusions

In conclusion, these results show that semen dilution with AMSC-EVs can modulate endometrial inflammation induced by insemination without altering sperm motility parameters and suggest that this approach might have the potential to reduce the risk of PPBIE in susceptible mares post-insemination.

This work has laid the foundation to better clarify the mechanisms by which AMSC-EVs can downregulate the uterine inflammatory response.

## Figures and Tables

**Figure 1 ijms-24-05166-f001:**
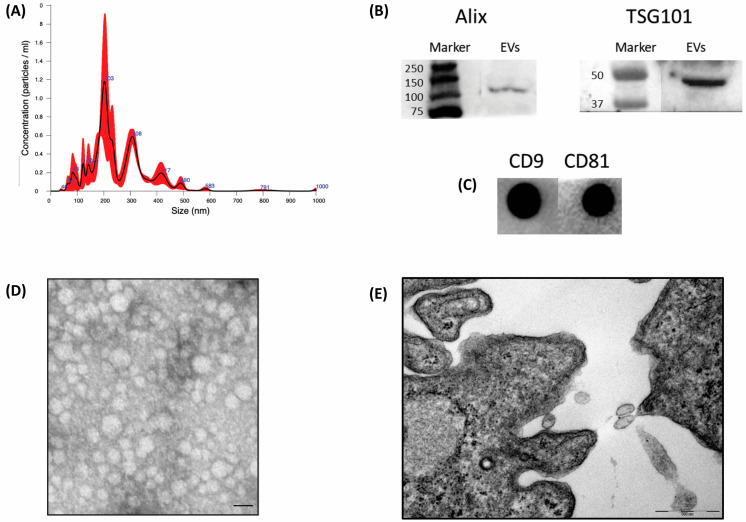
Characterization of amniotic EVs. (**A**) Results of the NanoSight analysis; (**B**) Western blot for internal EV marker and contaminant (Alix and TSG101); (**C**) dot blot for membrane markers (CD9 and CD81); (**D**) transmission electron microscopy analysis of EVs inside the seminal plasma showing typical morphological characteristics of vesicles (scale bar: 0.2 μm); (**E**) the ultrastructural observation of a mesenchymal stromal amniotic cell, allowing us to highlight the presence of extracellular membranous vesicles with electron-lucent or slightly electron-dense content located near the cells of origin (scale bar 500 nm).

**Figure 2 ijms-24-05166-f002:**
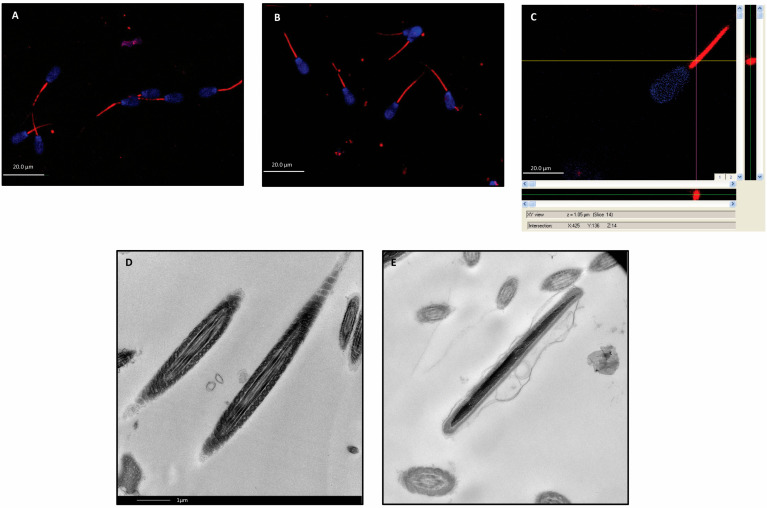
Dose–response curve. (**A**) At 400 × 10^6^ EVs/mL, the incorporation was uniform. (**B**) Incorporation with red dots at lower concentration. (**C**) z-Stack image obtained via confocal microscopy. EVs are included within the tail and not resting on the membrane. (**D**) Electron microscope figure of control spermatozoa incubated in absence of EVs. (**E**) Electron microscope figure of spermatozoa incubated with EVs, which are incorporated in the middle tract. This figure shows an intact sperm membrane (scale bar 1 µm).

**Figure 3 ijms-24-05166-f003:**
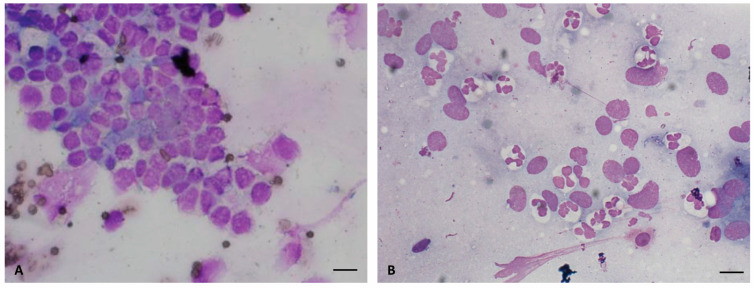
Endometrial cytology stained with dip quick stain (×100). (**A**) Negative endometrial cytology with absence of inflammatory cells. (**B**) Positive endometrial cytology with few endometrial epithelial cells and hypersegmented neutrophils (scale bars 20 μm).

**Figure 4 ijms-24-05166-f004:**
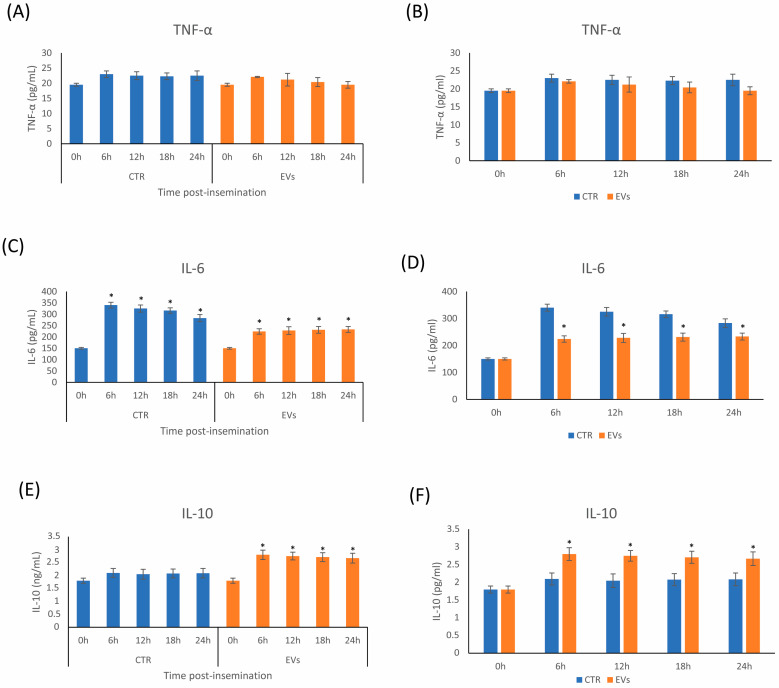
Concentrations of cytokines TNF-α (**A**), IL-6 (**B**) and IL-10 (**C**) in the uterine fluid of mares susceptible to persistent breeding-induced endometritis. CTR (control) are mares without intrauterine treatment; EV group are mares inseminated with semen supplemented with amniotic-derived EVs. Asterisks denote the effects during time periods (6–12–18–24) compared to time 0 for each experimental condition (*p* < 0.05). In (**D**–**F**), asterisks compare the effects during time periods (6–12–18–24) of EV group compared to CTR group (*p* < 0.05).

**Table 1 ijms-24-05166-t001:** Total motility and progressive motility concentration of sperm treated at different times with different concentrations of EVs, compared to CTR.

EV Concentration	Motile (%)					Progressive (%)				
(×10^6^)/mL	0 h	1 h	2 h	3 h	4 h	0 h	1 h	2 h	3 h	4 h
CTR	38.2 ± 1.87	32.9 ± 1.43 ^aA^	19.1 ± 0.43 ^aB^	19.0 ± 0.76 ^aB^	13.9 ± 1.06 ^aC^	8.8 ± 0.65	8.3 ± 0.88 ^aA^	6.4 ± 0.86 ^aB^	6.3 ± 0.72 ^aB^	2.5 ± 0.75 ^aC^
50		28.5 ± 1.32 ^bA^	18.7 ± 0.68 ^aB^	19.6 ± 0.98 ^aB^	10.6 ± 0.87 ^bC^		10.7 ± 0.76 ^bA^	6.3 ± 0.75 ^aB^	6.8 ± 0.71 ^aB^	3.5 ± 0.54 ^bC^
100		27.9 ± 1.29 ^bA^	17.8 ± 0.54 ^aB^	24.7 ± 0.86 ^bC^	14.1 ± 0.76 ^aD^		10.3 ± 0.86 ^bA^	9.1 ± 0.99 ^bB^	7.3 ± 0.20 ^aC^	3.1 ± 0.42 ^bD^
150		27.3 ± 1.93 ^bA^	18.6 ± 0.78 ^aB^	27.5 ± 0.65 ^cA^	15.0 ± 0.18 ^aC^		9.3 ± 0.53 ^cA^	9.8 ± 0.12 ^bA^	7.5 ± 0.32 ^aB^	3.1 ± 0.87 ^bB^
200		23.3 ± 1.12 ^cA^	18.6 ± 0.52 ^aB^	24.4 ± 0.29 ^bA^	18.7 ± 0.85 ^cB^		8.2 ± 0.39 ^cA^	9.0 ± 0.23 ^bB^	7.3 ± 0.73 ^aA^	3.3 ± 0.50 ^bC^
250		23.1 ± 1.24 ^cA^	16.4 ± 0.49 ^bB^	25.6 ± 0.33 ^bA^	18.1 ± 0.97 ^cB^		7.7 ± 0.79 ^cA^	9.2 ± 0.21 ^bB^	7.4 ± 0.77 ^aA^	3.2 ± 0.63 ^bC^
300		27.4 ± 1.52 ^bA^	12.5 ± 0. 33 ^bB^	22.2 ± 0.29 ^dC^	17.0 ± 0.67 ^cD^		10.3 ± 1.06 ^bA^	9.6 ± 0.34 ^bA^	7.9 ± 0.45 ^aB^	3.4 ± 0.11 ^bC^
350		27.3 ± 1.69 ^bA^	18.2 ± 0.55 ^aB^	25.4 ± 0.26 ^bA^	17.1 ± 0.88 ^cB^		12.7 ± 0.98 ^dA^	9.8 ± 0.43 ^bB^	8.4 ± 0.18 ^bC^	3.2 ± 0.42 ^bD^
400		33.0 ± 1.63 ^aA^	26.3 ± 0.74 ^cB^	23.8 ± 0.99 ^bC^	23.9 ± 1.09 ^dC^		12.1 ± 0.77 ^dA^	9.1 ± 0.66 ^bB^	8.0 ± 0.21 ^bC^	6.0 ± 0.74 ^cD^
450		25.3 ± 1.02 ^cA^	26.3 ± 0.44 ^cA^	15.1 ± 0.54 ^eB^	21.3 ± 1.06 ^dC^		12.2 ± 1.03 ^dA^	8.3 ± 0.74 ^bB^	5.7 ± 0.63 ^cC^	6.3 ± 0.83 ^cC^
500		12.5 ± 1.00 ^dA^	31.9 ± 0.2 ^dB^	21.4 ± 0.77 ^dC^	16.9 ± 1.98 ^cD^		11.1 ± 0.94 ^dA^	8.3 ± 0.79 ^bB^	5.4 ± 0.66 ^cC^	6.0 ± 0.49 ^cC^

Legend: EV: amniotic derived extracellular vesicle; CTR: control. Different capital letters (A–D) in superscript indicate statistically significant differences (*p* < 0.05) between 0 and 4 h incubation times. Different lowercase letters (a–d) in superscript indicate statistically significant differences (*p* < 0.05) between different EV concentrations.

**Table 2 ijms-24-05166-t002:** Sperm parameters of the stallion used for in vivo study.

Parameters	Values
Spermatozoa concentration/mL	146.54 × 10^6^
Volume	15 mL
pH	7.6
Motility	51.8% of which 34.4% with progressivity
Vitality	82%
Morphology	80% normal spermatozoa
Primary anomalies	4%
Secondary anomalies	16%

**Table 3 ijms-24-05166-t003:** Percentage of polymorphonuclear neutrophils in treatment and control groups.

	0 h	6 h	24 h
CTR	2%	80% ^aA^	20% ^bA^
EVs	2%	45% ^aB^	12% ^bB^

Legend: CTR: control; EVs: extracellular vesicles derived from amniotic cells. Lowercase letters (a,b) in superscript indicate statistically significant differences (*p* < 0.05) within the same group. Capital letters (A,B) in superscript represent statistically significant differences (*p* < 0.05) between the two experimental groups.

**Table 4 ijms-24-05166-t004:** Value in mm^2^ of intrauterine fluid evaluated in the control and treated mares.

	0 h	6 h	24 h
CTR	100	1000 ^aA^	500 ^aA^
EVs	100	150 ^aB^	130 ^bB^

Legend: CTR: control; EVs: extracellular vesicles derived from amniotic cells. Lowercase letters (a,b) in superscript indicate statistically significant differences (*p* < 0.05) within the same group. Capital letters (A,B) in superscript represent statistically significant differences (*p* < 0.05) between the two experimental groups.

## Data Availability

The datasets used and analyzed are available from the corresponding author on reasonable request.

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
