# Peer review of "Amniotic Mesenchymal-Derived Extracellular Vesicles and Their Role in the Prevention of Persistent Post-Breeding Induced Endometritis"

_ijms, 2023, doi:10.3390/ijms24065166_

Round 1
Reviewer 1 Report
The manuscript “Amniotic mesenchymal derived extracellular vesicles and their role in the prevention of persistent post-breeding induced endometritis” brings exciting results. However, some concerns should be addressed.
The main concerns regarding the manuscript are related to the experimental design.
How were the EVs extended before freezing, and how were they thawed and processed before dilution into semen? Could you provide the osmolarity and pH?
Why was semen diluted to 10 million sperm per mL in the in vitro study?
Also, why wasn’t the same sperm dilution used in the in vivo study? Was the stallion used in the in vivo study one of the stallions used for the in vitro experiment?
Why didn’t the authors access sperm plasma membrane integrity after incubation with EVs? It should also be discussed since it was speculated and shown that there was an incorporation of EV by the spermatozoa.
The main concern about the experiment is that the in vivo experiment wasn’t a crossover design study, which might influence the results since the number of mares enrolled is limited.
L543-572 is confusing and should be rewritten. First, the authors mentioned that “(L550)semen was diluted to obtain 500x106 vital spermatozoa with progressive motility in 10-20 ml.”, then after that, “(L567) The insemination dose was set at 1x109 total spermatozoa, diluted in 30 ml of INRA96 and supplemented with 400x108 AMC-EVs, considering 400x106 vesicles per 10x106/ml of semen.” – Please, review and both sections could be merged for clarification.
The characterization of the susceptibility to endometritis should be clarified and expanded.
Why was endometrial cytology not collected also at 48/72 h after breeding?
L586 – “Mares were treated with uterine lavages with 2 L of Ringer solution 24 hours after the insemination, immediately after sample collections and after oxytocin administration (20 units, i.m.), twice a day until 96 hours post-insemination.” – this sentence is confusing. Please, clarify. Were uterine lavages performed every time after sampling and after oxytocin?
Why were the concentrations of cytokines not evaluated in the EVs?
Minor concerns
Abstract
L18 – PBIE is the persistent/delayed uterine inflammation, not the result of persistent inflammation. Please, review.
Avoid repeating words already used in the Title as keywords. – Please review.
L49 and 53 – Although it’s known that IL6 has both pro- and anti-inflammatory effects, the paragraph is confusing for the reader. Please, review.
L64-66 – Full papers about PRP therapies in mares have been missed. Please, include.
Results
All questions regarding the methodology should address the results in this section.
Why the sperm motility parameters were so low if the stallions used in the study had proven fertility?
Tables 2A,B,C can be a supplement.
Discussion
The discussion is missing all the limitations of the study. Please, include
Could the pre-breeding intrauterine infusion of EVs also be an alternative treatment? Please, discuss.
The authors are missing important information about pre-breeding cytokine expression in susceptible mares and some pathways associated with post-breeding uterine inflammation that could improve the paper's discussion.
Reviewer 2 Report
The manuscript described that stallion semen supplemented with extracellular vesicles derived from amniotic mesenchymal cells (AMC-EVs) decreased TNF-α and IL-6 levels and increased anti-inflammatory IL-10 level, thus preventing the development of PPBIE in mare. There are many problems to be addressed.
1. What is the basis for the selection of susceptible mares in the study?
2. The authors stated "Mare placentas (n=3) were collected from three broodmares at term of a normal pregnancy". How about the parity of these three mares? When was the last production? Are they healthy? Do researchers have access to production data?
3. Where is the identification of amniotic mesenchymal stem cells (AMCs)? The pictures of surface marker identification mentioned in the method must be provided.
4. Did AMC refer to mesenchymal cells or mesenchymal stem cells? The descriptions are inconsistent.
5. Extracellar vesicles is a group of mixtures, including exosomes, MVs and apoptotic bodies. The authors should identify the type of extracellar vesicles used in this study.
6. Identification of EVs markers should be compared with cell cultures and cell fragments
7. The picture of TEM is too terrible to observe the existence of EVs.
8. The cytokine levels of the control group and the treatment group should be compared.
9. Where are the microscope pictures used to analyze PMN?
10. The English writing of the whole article needs to be carefully checked and revised. For example, "citobrush" in line 412 should be "cytobrush"
Round 2
Reviewer 1 Report
The authors made a significant improvement in the manuscript.
Some important points should be addressed before the publication:
As mentioned before, there is a difference in the concentration of EV/sperm between the in vitro and in vivo studies. Although the amount of EV per mL is the same, sperm dilution is not. It seems that it must be discussed. In the in vitro study, there were 400 million EVs per 10 million sperm, whereas, in the in vivo study, there were 400 million EVs per 50 (500 in 20 mL) to 25 (500 in 10 mL) million viable sperm.
Could the sperm parameters from the semen used for AI be provided? Similarly, as mentioned before, the sperm parameters in the in vitro study are pretty low. The authors should discuss it.
Add the lack of crossover as a limitation of the study. The authors could mention that the mares were privately owned mares.
Line 337 - Please review "wich" should be ", which" "Leukocytes than release PG wich promote myometrial contractility [62]."
Line 338 - 342 - It's confusing - please rewrite
